



# Measurement of Black Carbon Emissions from Multiple Engine and Source Types using Laser-Induced Incandescence: Sensitivity to Laser Fluence

Ruoyang Yuan[1], Prem Lobo[2], Greg J. Smallwood[2], Mark P. Johnson[3], Matthew C. Parker[3], Daniel Butcher[4], Adrian Spencer[4]

[1]Department of Mechanical Engineering, University of Sheffield, Sheffield, S1 3JD, United Kingdom
[2]Metrology Research Centre, National Research Council Canada, Ottawa, Ontario, K1A 0R6, Canada
[3]Rolls-Royce plc, Derby, DE24 8BJ, United Kingdom
[4]Department of Aeronautical and Automotive Engineering, Loughborough University, Loughborough, LE11 3TU, United Kingdom

*Correspondence to*: Ruoyang Yuan (ruoyang.yuan@sheffield.ac.uk)

**Abstract.** A new regulatory standard for non-volatile particulate matter (nvPM) mass concentration emissions from aircraft engines has been adopted by the International Civil Aviation Organisation. One of the instruments used for the regulatory nvPM mass emissions measurements in aircraft engine certification tests is the Artium Technologies LII 300, which is based on laser-induced incandescence. The LII 300 has been shown in some cases to demonstrate a variation in response to the type of black carbon particle measured. Hence it is important to identify a suitable black carbon emission source for instrument calibration. In this study, the relationship between the nvPM emissions produced by different engine sources and the response of the LII 300 instrument utilising auto-compensating laser-induced incandescence (AC-LII) method was investigated. Six different sources were used, including a turboshaft helicopter engine, a diesel generator, an intermediate pressure test rig of a single sector combustor, an auxiliary power unit gas turbine engine, a medium-sized diesel engine, and a downsized turbocharged direct injection gasoline engine. Optimum LII 300 laser fluence levels were determined for each source and operating condition evaluated. It was found that an optimised laser fluence can be valid for real-time measurements from a variety of sources, where the mass concentration was independent of laser fluence levels covering the typical operating ranges for the various sources. However, it is important to perform laser fluence sweeps to determine the optimum fluence range, as differences were observed in the laser fluence required, between sources and fuels. We discuss the measurement merits, variability, and best practices in the real-time quantification of nvPM mass concentration using the LII 300 instrument, and compare that with other diagnostic techniques, namely absorption–based methods such as photoacoustic spectroscopy using a photoacoustic extinctiometer (PAX) and a Micro Soot Sensor (MSS), and thermal-optical analysis (TOA). Particle size distributions were also measured using a scanning mobility particle sizer (SMPS). Overall, the LII 300 provides robust and consistent results when compared with the other diagnostic techniques across multiple engine sources and fuels. The results



from this study will inform the development of updated calibration protocols to ensure repeatable and reproducible measurements of nvPM mass emissions from aircraft engines using the LII 300.

## Copyright statement

The works published in this journal are distributed under the Creative Commons Attribution 4.0 License. This licence does not affect the Crown copyright work, which is re-usable under the Open Government Licence (OGL). The Creative Commons Attribution 4.0 License and the OGL are interoperable and do not conflict with, reduce or limit each other.

© Crown copyright 2021

## 1 Introduction

Short- and long-term exposure to particulate matter (PM) can lead to serious health problems such as lung or heart disease (AQEQ, 2005). There is increasing emphasis on reducing PM emissions from energy conversion systems especially for the transport sector. World Health Organisation (WHO) reported that up to 50% of PM emissions in OECD countries were caused by road transport, in which diesel traffic was the majority (United Nations Environment Programme World Health Organisation, 2009). In addition to road traffic, aviation and shipping are also significant sources of PM emissions in the transport sector. Therefore, it is strategically important to limit the exposure to PM from various sources, and accurate measurement of PM is a key enabler to achieving this goal and developing low PM emission transport solutions.

The International Civil Aviation Organisation (ICAO) Committee on Aviation Environmental Protection (CAEP) has adopted new Standards and Recommended Practices (SARPs) limiting aircraft engine non-volatile particulate matter (nvPM) (also referred to as soot or black carbon) number and mass emissions to mitigate the impact of aircraft engine emissions on local air quality (ICAO, 2017). The nvPM refers to the particles that exist at the aircraft engine exhaust nozzle exit plane that do not volatilise at temperatures greater than 350°C (ICAO, 2017). A standardised sampling and measurement methodology for aircraft engine nvPM emissions has been developed by the Society of Automotive Engineers (SAE) which has been detailed in Aerospace Information Report (AIR) 6241 (SAE, 2013), Aerospace Recommended Practice (ARP) 6320 (SAE, 2018) and adopted by ICAO (ICAO, 2017). The SAE and ICAO documents also define the performance criteria and calibration protocols for the instruments to be used in the standardised measurements. Currently, the two instruments that satisfy the performance criteria for nvPM mass measurements are the Artium Technologies LII 300, based on laser-induced incandescence (LII), and the AVL Micro Soot Sensor (MSS), based on photoacoustic spectroscopy. The standardised systems for aircraft engine nvPM emissions measurements using these instruments have been previously evaluated and inter-compared (Lobo et al., 2015a, 2020).

The instruments used to measure nvPM mass concentration in the exhaust of aircraft engines are calibrated by reference to elemental carbon (EC) content, as determined by thermal optical analysis (TOA) of a diffusion flame combustion aerosol source (Durdina et al., 2016). TOA is an industry accepted method for measuring the mass of both organic carbon (OC) and





elemental carbon (EC) aerosols sampled on a quartz filter (Baumgardner et al., 2012). The only criteria for the source used for

calibrating nvPM mass instruments is that it be capable of producing an average EC content of the collected mass on the filter of ≥80% (SAE, 2018).

The LII techniques have been shown to be sensitive to different black carbon types, both for continuous wave laser LII (Baumgardner et al., 2012; Gysel et al., 2012; Laborde et al., 2012) and for pulsed laser LII as used in the LII 300 (Durdina et al., 2016). The relationship between the nvPM emissions produced by these different sources and the response of the LII

300 instruments has not been fully investigated. It is important to identify a suitable nvPM emissions source that meets the requirements to provide reliable and robust calibration of the LII 300 instrument used in the emissions certification of aircraft engines.

The LII technique has been widely applied in combustion studies to obtain in-situ black carbon characteristics, such as soot volume fraction (Boiarciuc et al., 2006; Choi and Jensen, 1998; Liu et al., 2011; Melton, 1984; Quay et al., 1994; Zhang et al.,

2019) and primary particle size measurements (Axelsson et al., 2000; Boiarciuc et al., 2006; Boies et al., 2015; Mewes and Seitzman, 1997), from both laboratory flames (Axelsson et al., 2000; Mewes and Seitzman, 1997; Tian et al., 2017; Zhang et al., 2019) and aircraft engines (soot concentration: Schäfer et al., 2000; Delhay et al., 2009; Black and Johnson, 2010; Petzold et al., 2011; Lobo et al., 2015a; and soot particle sizes: Boies et al., 2015). Previous reviews on LII have provided insight on current challenges, questions associated with this technique (Schulz et al., 2006), and current model developments (Michelsen

et al., 2015). Several models (Eckbreth, 1977; Kock et al., 2006; Lehre et al., 2003; Melton, 1984; Michelsen et al., 2007; Roth and Filippov, 1996; Schittkowski et al., 2002; Smallwood et al., 2001; Snelling et al., 2004) of the LII technique have been reviewed to quantify the soot concentration and primary particle size. These were required because the complex mechanisms incorporated in the LII model strongly influenced the predicted signal magnitudes and time-evolution. The signal dependence on the laser fluence, and in turn the quantification of soot concentration, was highlighted in each model. As the laser fluence

increases, soot absorbs increasing amounts of light energy, resulting in greater particle peak temperature and radiation magnitude. The subsequent decay rate of this radiation is related to the particle temperature, which in turn is related to the primary particle size. Sublimation can also occur at excessively high laser fluence levels, characterised by the particle-volume reduction. According to the model by Michelsen et al., 2003, the maximum LII signal may level off or be reduced due to this mass loss. Although there are differences in the temporal profiles, the various models provide similar results for the relative

signal magnitudes as a function of laser fluence (Schulz et al., 2006). In conventional LII techniques, the determination of the LII fluence level directly affects the accuracy of the quantitative measure of the soot concentration. Additionally, further calibration (via laser extinction for example) is required to obtain absolute rather than relative measurements. The requirement to accurately represent the target nvPM absorption and sublimation characteristic during calibration is an additional complexity associated with the LII technique. An improved approach, described in the literature (Snelling et al., 2005; Thomson et al.,

2006), utilises an auto-compensating laser-induced incandescence (AC-LII) method. By measuring LII signals at two separate wavelengths and temporally resolved decay rates, AC-LII has the potential to resolve the soot temperature, volume fraction,



and particle size without the need for additional calibration sources. This method was incorporated into the commercial LII 300 instrument by Artium Technologies Inc.

The goal of this study is to assess the suitability of the nvPM emissions from a variety of different engines and fuels to be

used as a calibration source for the LII 300 instrument, for application to the emissions certification of aircraft engines. It was important to identify a source that met all the requirements, produced a similar response from the LII 300 to that obtained from nvPM emissions from an aircraft gas turbine engine, and one that was economical and practical to operate. In this study, we evaluated a range of different combustion sources that could potentially be used as calibration sources, including a turboshaft helicopter engine (already verified as an applicable aircraft engine calibration source for LII 300), an intermediate-pressure

single injector combustor test rig, an auxiliary power unit gas turbine engine, a diesel generator, a medium-sized diesel engine, and a downsized turbocharged direct injection gasoline engine, which utilised different fuel types: kerosene, diesel, and gasoline.  The relationship between the nvPM emissions produced by the different engine sources and the response of the LII 300 instrument was investigated. Laser fluence sweeps were performed at different operating conditions, and the fluence dependence and variability of the technique for field measurements of different nvPM sources were studied. The measurements

using the LII 300 were compared with other diagnostic techniques, including thermal optical analysis and photoacoustic spectroscopy, for comparison. The relationship between laser fluence and measurement performance is discussed. The results from this study will help identify nvPM sources that can be used to calibrate the LII 300 instrument, and inform the development of updated calibration protocols to ensure repeatable and reproducible measurements of nvPM mass emissions from aircraft engines using the LII 300, including a procedure to optimise the laser fluence.

**2. Experimental Method**

**2.1 Diagnostics methods**

2.1.1 Laser-Induced Incandescence (LII)

Laser-induced incandescence (LII) measures the thermal emission from soot particles heated by a laser to temperatures in the 2500 - 4500 K range (Bachalo et al., 2002). Assuming that all the volatile and semi volatile OC that condense on the BC

particles will be evaporated promptly at these temperatures, the LII signals are directly related to the non-volatile particles, primarily refractory black carbon (rBC), a form of carbon directly related to EC (Baumgardner et al., 2012). The LII 300 instrument calculates a mass concentration from the particle emission signal and the temperature determined from the signal at two wavelengths using two-colour pyrometry.  As the optical properties of the particle may vary from source to source, a correlation factor may need to be applied to this optical determination of the mass concentration to relate the result to that of

EC determined from TOA. A multimode pulsed Nd:YAG laser, operating at fundamental wavelength of 1064 nm, with a pulse duration of 7 ns full width half maximum (FWHM) and a repetition rate of 20 Hz was used as the excitation source. The laser





beam passed through a set of optics and formed into a square cross-section light beam uniformly incident throughout the probe volume. The measurement volume is approximate 14.7 mm$^3$ and the range of fluence levels is typically 0.6-3.2 mJ/mm$^2$.

At low laser fluence levels, the peak LII signal intensity rises with the laser fluence as peak particle temperatures increase.
As laser fluence is increased, a threshold is reached where the measured mass concentration becomes independent of laser fluence. This plateau region extends for a range of increasing laser fluence until level is reached where soot sublimation begins to dominate. At these very high laser fluences, the mass loss associated with soot sublimation has an influence and a reduction in the mass concentration may be observed (Michelsen, 2015). Various soot fragments evaporate from the particle surface when sublimation occurs (Schulz et al., 2006). Models usually assume these to range from C1 to C7 (molecules having 1 to 7
carbon atoms) dependent on the particle temperature (Leider et al., 1973). Additional research is required to understand the sublimation loss which is complicated by uncertainties in physical parameters of the carbon species, limiting the ability of the models to fully predict phenomena in this regime (Michelsen, 2015).

In the AC-LII technique (Snelling et al., 2005), the soot temperatures are determined by the two-colour pyrometry method (in the LII 300, narrow bandpass filters centred at ~446 nm and ~720 nm are used with bandwidths of 40 and 20 nm
respectively). The peak soot temperature and absolute intensity of the LII signal were used to calculate soot volume fraction ($f_v$). With soot particle material density ($\rho_s$) from literature, the mass concentrations of the non-volatile particles, or EC are obtained from $f_v \times \rho_s$. The primary particle sizes are determined from the temperature decay rates with the assumption that conduction dominates the particle cooling rates for conditions below the sublimation limit (Smallwood, 2008).

Theoretically, the mass concentration obtained from the two-colour method is valid regardless of the laser fluence applied,
and a lower laser fluence level might be preferred to avoid the reduction in particle volume (sublimation) at high laser fluence. Practically, care should be taken with low laser fluence levels for particles from unknown sources with unknown combination of size distributions and morphologies, as the assumption of the uniform temperature in the sampling volume could be invalid (Liu et al., 2016). The particles with various sizes could reach different peak temperatures at separate times, which induce uncertainties with the determination on the effective peak particle temperatures, and in turn uncertainties with the derived mass
concentration values. For a typical effective soot temperature of 4000 K, a 100 K error (2.5% of the soot temperature) can lead to 15% error on the estimated soot volume fraction value (Boiarciuc et al., 2006). It is important to operate the LII 300 with optimum laser fluence, in a region of laser fluence where the reported mass concentration is relatively independent of fluence, to minimise uncertainties. In the current work, laser fluence sweep experiments (varying laser fluence from low levels to high fluence conditions with all other parameters held constant) were performed for each source at different operating conditions.
This was performed to examine the relationship between the measured mass concentrations and LII fluence levels for multiple sources, and to subsequently determine the optimum laser fluence level for both calibration and application to real-time measurements from aircraft engines.



### 2.1.2 Thermal Optical Analysis (TOA)

TOA measures the mass of both organic carbon (OC) and elemental carbon (EC) of the particulate matter sampled on a quartz

filter. This is not a gravimetric measurement where the mass of the elemental carbon was measured, but rather a chemical analysis of the black carbon on the filter. The NIOSH 5040 (NIOSH, 2003) standard method was used for the TOA, with a modified temperature ramp as specified by ARP 6320 (SAE, 2018), to obtain the EC mass concentrations. The uncertainty associated with this method for EC mass concentration is stated to be 16.7% (NIOSH, 2003), due to additional uncertainty in assigning the split between OC and EC.  The uncertainty associated with the total carbon (TC) mass concentration is 8.4%

(Conrad, 2019). A basic overview of the process is presented here. The sample flow was pulled across a mounted clean quartz filter with the total volume of the sample flow measured. The particulate matter (mostly black carbon) in the sample flow was captured by the filter. After collection, a 1cm × 1cm punch from the filter was cut out and mounted in an analyser (Sunset Laboratory OC-EC Aerosol Analyser) to obtain the EC and OC content. The filter punch was heated in the analyser oven in stages. From 0 to 425 s, the sample was surrounded by an inert helium atmosphere and the heating process drove off any

embedded volatile organic compounds (VOCs) from the sample. The desorbed VOCs were further drawn in to a series of catalyst beds where they were converted to $CO_2$ and then to methane, whose concentration was measured by a flame ionisation detector (FID) to give a measure of the organic carbon fraction on the filter punch. As the VOCs are heated, it is possible that some were pyrolysed and converted to char, which stayed on the filter punch, instead of being driven off the surface. The quantity of the char was measured via laser extinction propagated through the sampled filter punch. From 425 to 800 s, the

sample was surrounded by a reactive oxygen/helium atmosphere. As the oven heated the sample the elemental carbon reacted with the oxygen to form $CO_2$, which was again converted to methane and measured by the FID. The elemental carbon fraction of black carbon on the filter punch was then measured by subtracting the char (which was originally organic carbon) content from the total. With the two mass values quantified, i.e. the OC mass fraction and the EC mass fraction, the total carbon (TC) on the filter punch can be obtained. By multiplying by the stain area of the filter, the total mass of EC, OC and TC on the entire

filter were quantified. Having also measured the total volume of sampled gas, the mass concentration of the EC, OC and TC were obtained. One uncertainty of the OC quantification from the TOA measurements was the artefact effect caused by the gas-phase semi-volatile organic compounds absorbed on the filter (Durdina et al., 2016). To correct this artefact, the OC determined from a secondary filter that contained exclusively absorbed gas-phase OC, was subtracted from the OC mass determined from the primary filter.


### 2.1.3 Photoacoustic Spectroscopy

In this study, a Photoacoustic Extinctiometer (PAX, Droplet Measurement Technologies, Inc.) and a Micro Soot Sensor (MSS, AVL, model 483) were used to provide additional measurements for nvPM mass concentrations. Both the PAX and MSS are based on the photoacoustic method (Adams et al., 1989), measuring the acoustic wave generated by the heated gases



surrounding the light absorbing particles due to their increased temperature after interaction with a laser. Unlike filter-based absorption methods, where uncertainties related to organic aerosol coating and light absorbing properties are issues (Baumgardner et al., 2012; Corbin et al., 2019), photoacoustic measurements can potentially be affected by the humidity via latent heat and mass transfer from volatile droplets (Arnott, 2003) and absorption by molecular species present in the gaseous medium. The PAX and the MSS measurements rely on the light absorption properties of the PM sources, and for which a mass absorption cross section (MAC) is usually assumed. In this work, a MAC of $7.5 \pm 1.2$ $m^2g^{-1}$ at the wavelength of 550 nm was applied to the PAX following the recommendation by Bond et al. (Bond and Bergstrom, 2006), noting that the MAC will vary with the black carbon source and operating condition. The MAC for the PAX (with a measured absorption at a wavelength $\lambda$ of 870 nm) were converted using: $MAC(\lambda) = MAC(550 \text{ nm}) \times (550 \text{ nm}/\lambda)$, assuming the refractive index was the same at those wavelengths, equivalent to stating that the aerosol Angstrom exponent (AAE) is 1. The MSS was previously calibrated with a soot source of known concentration (determined by comparison with EC from TOA).

### 2.1.4 Scanning Mobility Particle Sizing

Although the focus in this work was on the quantification of nvPM mass concentration, the particle size distributions from different engine sources were measured using a scanning mobility particle sizer (SMPS, TSI, USA). The SMPS consisted of an electrostatic classifier (model 3082), a differential mobility analyser (model 3081), a soft X-ray neutraliser (model 3088), and an ultrafine condensation particle counter (model 3776). The particle size distribution data of the nvPM emissions from the various sources provides additional information on the physical properties of the particles. This will also provide information for the LII model development in terms of particle size distribution functions. The typical size range for a SMPS scan was 7 – 206 nm.

### 2.2 Test rigs and fuels

Multiple sources (test rigs) were used to study the LII 300 instrument's response to the nvPM exhaust emissions at steady-state conditions. These rigs included an aircraft gas turbine turboshaft engine (Rolls-Royce Gnome helicopter engine, Rig A), an intermediate-pressure gas turbine combustor rig (Rolls-Royce IP rig, Rig B), a gas turbine auxiliary power unit (Rover 2S/150 APU, Rig C), a diesel generator (Stephill Generators SE6000D4, Rig D), a naturally aspirated medium-sized (4.4 litre) diesel engine (Rig E) and a downsized (1.0 litre) turbocharged gasoline direct injection (GDI) engine (Rig F). A schematic of the typical experimental setup for the different sources is shown in Figure 1. The source exhaust was diluted with HEPA-filtered air heated to 160°C. A cyclone with a 1μm cut size at a flow rate of 8 lpm was installed immediately downstream of the diluted engine exhaust. The output flow from the cyclone was transferred using a 3m long heated (100°C) carbon-loaded PTFE sample line, and split to a pair of ejector diluters (Dekati, Model DI-1000), operating in parallel to further dilute the sample. The flow from the ejector diluters was combined and transported through a mixing section to the inlet of a custom built sampling tunnel with 12 ports, which was used to distribute diluted exhaust sample, while ensuring there was excess flow. Carbon impregnated silicone tubing (nominal 3/8") were used to transfer sample to the real-time instruments and the





filter holders (for TOA, URG Corp. model URG-2000-30FVT). Quartz filter cassettes (model URG-2000-30FL) mounted in the filter holder were used to hold the quartz filters.

In this study, the focus was not on the performance of the test engine or the operating conditions, but rather the response of
the diagnostic instruments for a range of different combustion emission sources and associated fuel types - kerosene (Jet A-1) for the gas turbine engines and IP rig, diesel (EN590) for the diesel engines, and gasoline (EN228) for the GDI engine. The IP combustor rig design was developed as a tool for gas turbine fuel spray nozzle hardware ranking for soot production, similar to Makida et al. (2006). The soot consumption/oxidation process and maturity are different to that of typical rich-quench-lean gas turbine combustion. The real-time measurements of the nvPM emissions from these rigs using the LII 300 and the other
diagnostic instruments were recorded and compared. Table 1 summarises the operating conditions of the various test rigs. Fuel samples were acquired from each of the test rigs, and were subsequently analysed (Table 2). The Jet A-1 fuels used in test rigs A, B, C had similar properties, as did the EN590 fuels for test rigs D and E.

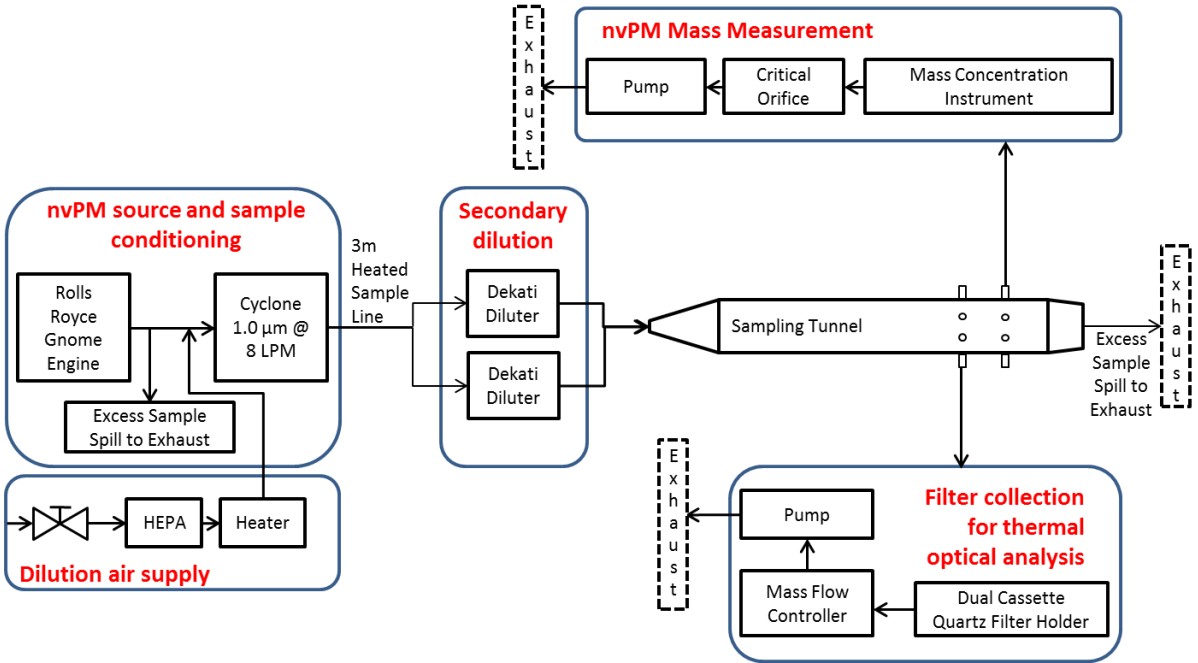


**Fig. 1**. Schematic diagram of the experimental setup.



**Table 1.** Test rigs and operating conditions

| Name | Source | Fuel type | Test conditions |
|---|---|---|---|
| Rig A | Gnome helicopter gas turbine engine | Kerosene (Jet A-1) | • Idle (11,500 rpm)<br>• High power output (HPO) (22,600 rpm)<br>• Pressure ratio 10:1 |
| Rig B | Intermediate-pressure combustor | Kerosene (Jet A-1) | • Low pressure low temperature (LPLT)<br>• Low pressure high temperature (LPHT)<br>• High pressure high temperature (HPHT)<br>• High pressure low temperature (HPLT)<br>• Pressure range 2 to 4 bar |
| Rig C | Auxiliary power unit gas turbine engine | Kerosene (Jet A-1) | • 6300 rpm<br>• Pressure ratio 3.76:1 |
| Rig D | Diesel generator | Diesel (EN590) | • 3 kW<br>• 5 kW |
| Rig E | Medium-sized diesel engine | Diesel (EN590) | • Load 1: Speed 1200 rpm, load 300 Nm<br>• Load 2: Speed 2200 rpm, load 165 Nm |
| Rig F | Downsized turbocharged gasoline direct injection engine | Gasoline (EN228) | • Speed 2500 rpm, load 100 Nm |


**Table 2a. Physical and chemical properties for the Jet A-1 fuels (Kerosene) used in Rigs A, B, and C**

| Property | Method | Allowable Range | Rig A | Rig B | Rig C |
|---|---|---|---|---|---|
| Specific Density, kg/m$^3$ (15 °C) | ASTM D4052 | 780-820 | 808.6 | 807.0 | 799.2 |
| Distillation temperature, °C | | | | | |
| - initial boiling point (I.B.P.) | IP 123 | | 150.2 | 148.2 | 153.9 |
| - 10% volume recovery at | IP 123 | 155-201 | 167.5 | 166.4 | 170.6 |
| - 50% vol. rec. at | IP 123 | | 198.0 | 196.0 | 197.8 |
| - 90% vol. rec. at | IP 123 | | 240.4 | 239.3 | 241.2 |
| - final boiling point (F.B.P.) | IP 123 | 235-285 | 258.0 | 257.0 | 268.7 |
| Aromatics, volume % | ASTM D1319 | 15-23 | 16.5 | 16.5 | 17.3 |
| Naphthalenes, volume % | ASTM D1840 | 0-3 | 1.86 | 1.73 | 1.59 |
| Smoke point, mm | ASTM D1322 | 20-28 | 22.3 | 22.5 | 23.6 |
| Hydrogen, mass % | ASTM D3343 | 13.4-14.3 | 13.72 | 13.74 | 13.86 |
| Sulphur, mass % | IP 336 | <0.3 | 0.033 | <0.030 | 0.070 |

**Table 2b. Physical and chemical properties for the EN590 fuels (Diesel) used in Rigs D and E**

| Property | Method | Rig D | Rig E |
|---|---|---|---|
| Derived Cetane Number | ASTM D6890 | 51.3 | 52.6 |
| Fatty Acid Methyl Esters, vol % | BS EN 14078 | 1.05 | 1.55 |
| Range | BS EN 14078 | A | A |
| Monocyclic Aromatics, mass % | IP 391 | 25.0 | 23.3 |
| Dicyclic Aromatics, mass % | IP 391 | 2.5 | 2.6 |
| Tri+ Aromatics, mass % | IP 391 | 0.1 | 0.2 |



| Polycyclic Aromatics, mass % | IP 391 | 2.6 | 2.8 |
|---|---|---|---|
| Aromatics, mass % | IP 391 | 27.6 | 26.1 |
| Smoke point, mm | ASTM D1322 | 16.9 | 17.6 |
| Hydrogen content, mass % | ASTM D5291 | 13.00 | 13.21 |

**Table 2c. Physical and chemical properties for the EN228 fuel (gasoline) used in Rig F**

| Property | Method | Rig F |
|---|---|---|
| Density at 15°C, kg/m$^3$ | ASTM D4052 | 775.2 |
| Distillation, °C | | |
| I.B.P. °C | ASTM D86 | 51.2 |
| F.B.P. °C | | 197.8 |
| Sulphur, mass % | ASTM D5453 | 0.0009 |
| Gross calorific value, MJ/kg | IP 12 | 44.08 |
| Vapour pressure at 37.8°C, kPa | ASTM D5191 | 33.6 |
| Hydrogen content, mass % | ASTM D3343 | 12.38 |
| Ethanol, volume % | EN ISO 22854 Proc. A | 2.20 |
| Methanol, volume % | | 0.03 |
| Naphthenes, volume % | | 6.6 |
| Olefins, volume % | | 2.6 |
| Aromatics, volume % | | 42.2 |
| Paraffins, volume % | | 43.5 |
| Oxygenates, volume % | | 5.03 |
| Benzene, volume % | | 0.85 |
| Saturates, volume % | | 50.1 |
| Oxygen content, mass % | | 1.29 |
| Methyl tert-butyl ether (MTBE) | | 2.68 |

## 3. Results and Discussion

### 3.1 LII 300 fluence optimisation

For real-time nvPM quantification with the LII 300, a single laser fluence level is typically used for the emission sources of interest. The choice of this laser fluence level plays an important role in maintaining the calibration of the instrument. For the experiments reported here, the laser fluence levels were deliberately tuned over a wide range, from low to sublimation levels of fluence, by adjusting the Q-switch ($Q_{sw}$) delay settings. Laser fluence sweep measurements (adjusting from low to high fluence with a number of discrete steps) were performed at steady-state engine operating conditions to determine the optimum fluence for a particular nvPM source and operating condition. For the laser fluence sweep tests at steady-state engine operation, measurements are unlikely be influenced by substantial variation in the source emissions. To account for any modest variations in the concentration of the source emissions, a time-weighted normalisation (TN) method was used to minimise the scatter caused by the variation from the source. This involved many repeats at a reference Q-switch delay during the fluence sweep and using the mean value at this reference Q-switch delay for initial normalisation of all data. Figure 2 shows an example of




the mass concentration time-series during one laser fluence sweep, while measuring the nvPM emissions from the IP gas turbine (Rig B). The red line denotes the reference points at which the Q-switch setting was fixed corresponding to the maximum laser energy output. The colourmap of data points shown in Fig 2(a) correlates to the Q-switch settings applied. Fig

2(b) is a time-series of the normalised mass concentration results from the time-weighted method. A best-fit was calculated on the TN-normalised mass concentrations using the local polynomial regression method (Loess method). The final normalisation was achieved by normalising the TN values to the peak value of the best-fit curve, as shown in Fig 2(c).

(a)                                                                        (b)


(c)

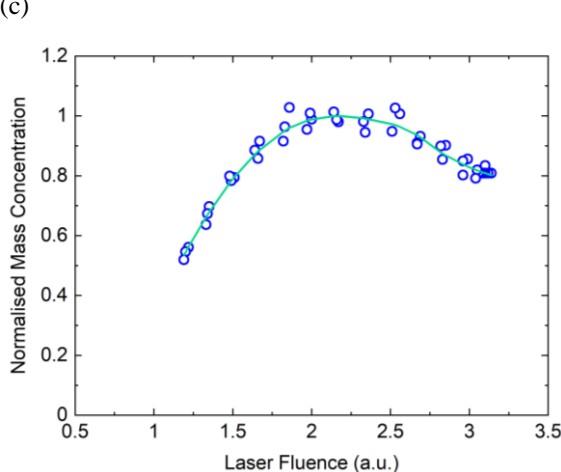

**Fig. 2.** Laser fluence optimisation procedure for mass concentration showing (a) all raw measurements, (b) initially normalised measurements using TN method, (c) normalised laser fluence sweep, superimposed with the best-fit curve (green) of the normalised mass concentration data using the Loess method with a second-order local polynominal regression.




To further examine the role of source emissions variability in the LII 300 fluence sweep measurements, the reference points of the 135$\mu$s Q-switch delay from the LII 300 were plotted together with the data simultaneously acquired by the PAX in Fig. 3. Results from the two sources (nvPM emissions from the IP rig (Rig B) and from the diesel engine (Rig E)) are presented.
Despite subtle differences in the variability of the two sources, the trends of the two signals between the PAX and the LII 300 are similar for both cases, suggesting that the uncertainties observed in the LII 300 fluence sweep measurements at the reference fluence points (red line in Fig. 2(a)) were likely influenced by the variability of the source. The trend also suggests engine thermal equilibrium was achieved after 600 s and there was ~ 2% variability in mass concentration in output following this. In the subsequent analysis, the TN method was used for the normalisation to minimise the impact of source variability.


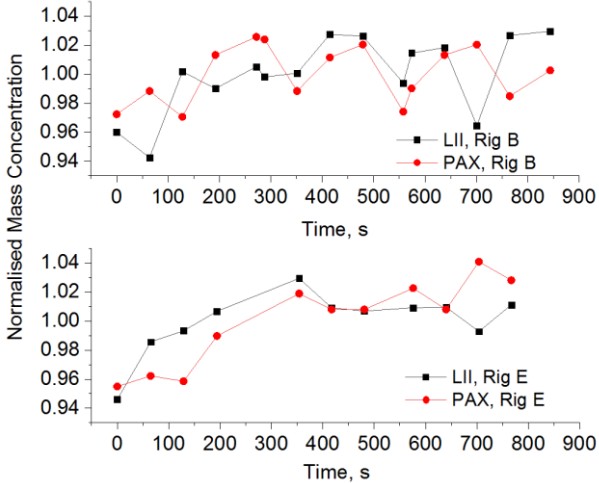

**Fig. 3.** Mass concentration (normalised) measurements by the PAX and the LII 300 at its reference laser fluence points during fluence sweep tests using two different sources: nvPM emissions from the IP combustor (rig B) and from the diesel engine (rig E).

### 3.2 Laser fluence dependence –Rig A

### 3.2.1 Profiles of the measured mass concentrations

Figure 4 shows the LII 300 laser fluence sweep vs normalised mass concentration (solid symbols) from the Gnome engine exhaust (Rig A) fueled with kerosene, at two different steady-state engine operating conditions – (1) idle and (2) high power output (HPO). It is anticipated that the idle and HPO conditions are two extremes in terms of the properties of the nvPM being
emitted from Rig A, with those at idle being less mature and with more volatile organic compounds, and the opposite for the HPO condition. Performing fluence sweep tests at these two conditions aided in determining an optimum laser fluence level



to cover the full range of conditions from this source for real-time nvPM mass measurements. At laser fluence levels below 1.7 (Fig. 4), the measurements showed an almost monotonic increase with fluence between the measured mass concentrations and the laser fluence. The measurement of nvPM mass concentration was independent of the laser fluence levels (nominal value) in the range of 1.9 to 2.5 for the HPO condition, and 2.2 to 3.2 for the idle condition. The shaded area in the Fig.4 corresponds to the data range that is within 2% of the peak value of the Loess best-fit curve[1] from the HPO condition. To determine an optimum laser fluence level for real-time nvPM mass concentration measurements, a fluence level at which the normalised mass concentration (NMC) data fell within the optimum range (within 2% error) at both conditions would be ideal, as these two engine conditions are anticipated to represent the extremes in terms of the properties of the nvPM emissions. For the Gnome engine, a fluence level at 2.4 (nominal, Fig. 4 arrow) was selected as optimum for this source across all operating conditions, and used in subsequent comparisons with results from the other diagnostic techniques (as discussed in section 3.5). While not the focus of this study, it is interesting to note the impact of laser fluence and source operating condition on the effective primary particle diameters (ePPD) resulting from the LII 300 measurements (via the decay rate of the LII signal) (Schulz, 2006). The relationship between the effective primary particle diameter and the laser fluence is most significant at low fluence levels (<1.9). As the reported sizes are larger than those reported for similar sources (Saffaripour et al., 2017), and as the ePPD is dependent on the rate of cooling of the particles, it is likely that there is an additional mechanism inhibiting the conduction of energy from the particles at low fluences. At high fluence levels (>2.5) sublimation is occurring and there may be a modest mass loss and potential reduction in particle size. In the region of optimum fluence the effective primary particle diameter is relatively insensitive to laser fluence. At all fluence levels, the effective primary particle diameters measured at idle were larger than those measured at the HPO condition. The obtained ePPDs at optimum fluence levels are within a similar range (6 – 19 nm by LII) reported for other gas turbine engines (Boies et al., 2015, Saffaripour et al., 2020). Further investigation into the ePPD measured by the LII 300 and corresponding size comparisons with other techniques such as TEM images will be addressed in future work, and is not discussed further as the focus of this study is on nvPM mass concentration measurements.

---

[1] The region near the peak of the Loess best-fit within the 2% error band is defined and referred as a plateau regime (relative uniform response to the fluence level) in the following sections. The optimum fluence discussed later is discussed in the plateau regime.





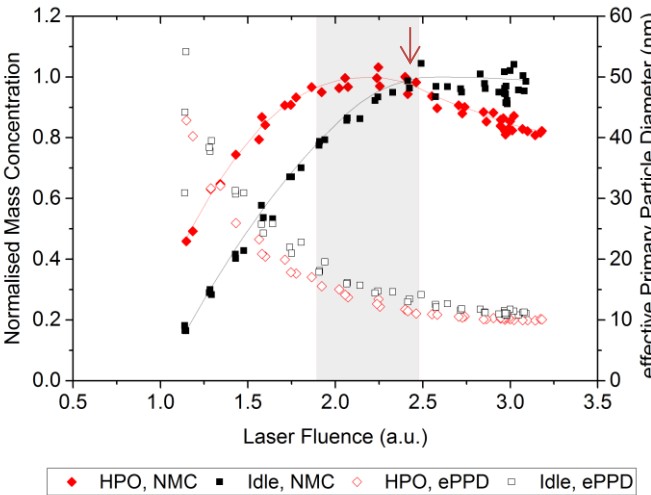

**Fig. 4.** Normalised mass concentration (NMC) mean and the derived effective primary particle diameter (ePPD) mean across the different laser fluence levels (nominal) using the nvPM source from the kerosene fuelled Rig A, at high power output (HPO) and at idle conditions (idle). Shaded area indicates the +/- 2% data range of the HPO case. Arrow indicates an optimum laser fluence level for real-time nvPM mass concentration measurements.

### 3.2.2 Sublimation

It was observed that at the HPO condition for Rig A, the mass concentration peaked at a laser fluence around 2.25, beyond which the mass concentration decreased moderately with increasing laser fluence, attributable to sublimation of the nvPM particles. However, at the idle condition, no obvious sublimation was observed over the range investigated for this laser fluence sweep experiment. The peak particle temperatures reported by the LII 300, shown in Fig. 5(a), permit further investigation into the difference in response to the nvPM produced at the idle and HPO conditions. The particles reach a lower peak temperature for the idle condition compared with that for HPO across the range of laser fluence levels applied. The combustion is fairly inefficient at the idle condition for Rig A, and >2000 ppm unburnt hydrocarbons have been previously observed at this condition. It is suspected that due to the combustion inefficiency at the idle condition, the soot particles measured at the engine exit plane were coated with VOCs (Olfert et al., 2017). To overcome this, additional laser energy was required to vaporise the VOCs, energy that was no longer available to heat the nvPM. To illustrate the potential for this phenomenon, the data obtained at the idle condition was aligned (via a shift by an arbitrary laser fluence) with the HPO condition and replotted (Fig. 5(b)), with the mass concentrations normalised as described previously. It was observed that the relationship between the normalised mass concentration and laser fluence exhibit very similar behavior at both the HPO and idle conditions. At both conditions, measured nvPM mass concentrations were found to be independent of laser fluence levels in the range 1.8 - 2.7, noting that the arbitrary scale for the idle condition has been shifted.





(a)

(b)

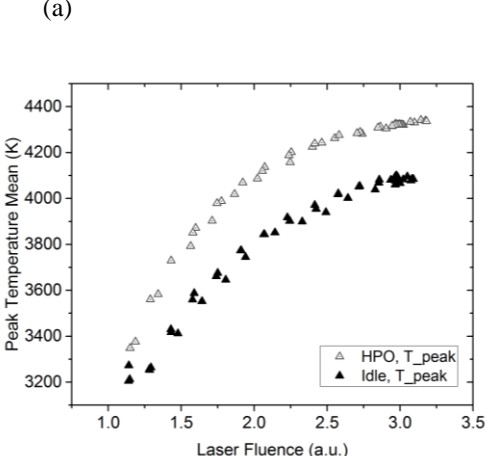

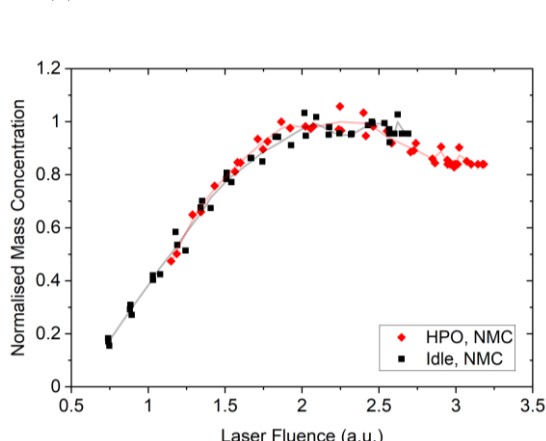

**Fig. 5.** (a) Peak particle temperature mean from the LII 300 for the range of laser fluence levels (nominal) at the two operating conditions. (b) Shifting of the data in Fig. 4 with the idle condition's data aligned with that for HPO. NMC - normalised mass concentration, using the nvPM source from the kerosene fueled Rig A.

### 3.2.3 Repeatability

The impact of the laser fluence on the mass concentration measurements shown in Figure 4 were also evaluated across multiple LII 300 instruments (LII 1 to LII 4) using the same nvPM source from Rig A to better understand instrument to instrument differences, and exclude potential single instrument anomalies in the observed trends with laser fluence levels. The results obtained from the HPO condition are plotted in Fig. 6(a). The trends of mass concentration measurements during the laser fluence sweep analysis across the four different LII 300 instruments were similar, with minor differences observed between the individual LII 300 instruments likely due to modest variations in the beam profiles from the lasers, the specifications of the optics, and the resulting shape and size of the laser sheet in the probe volume location. All the instruments demonstrated the existence of a plateau region, identified as the range of fluence levels where the mass concentration was independent of the fluence level applied. The fluence levels from each instrument were nominal values, as the energy meters in the instruments are not calibrated, although the nominal values are close to fluence units of mJ/mm$^2$. Without calibrated energy meters, the fluences reported for each instrument may not be directly compared to the others. To aid in interpretation of the results, fluence shifts were applied for different operating conditions (HPO, Idle in Fig. 5b), different instruments (LIIs 1 to 4 in Fig. 6b), and can be used for different rigs (Rigs A, C-F in Fig. 11b) to align the fluence sweep data. The data from LII 3 and LII 4 were shifted with an offset laser fluence value to fit with the data from LII 1 in Fig. 6(b). The data from LII 2 did not require shifting as the fit with the data from LII 1 was good. The normalised and shifted mass concentration profiles





from the four different LII 300 instruments were found to repeatably demonstrate similar behavior. While interpretation of
the behavior in the low fluence region (fluence <1.8 a.u. for Rig A) remains a topic to be explored, it is recommended that
measurements be acquired in the plateau region (1.8 < fluence < 2.7 a.u. for Rig A; Fig. 6(b)), and that extra care be taken in
interpreting the results from the low fluence and sublimation regions.

(a)                                                    (b)

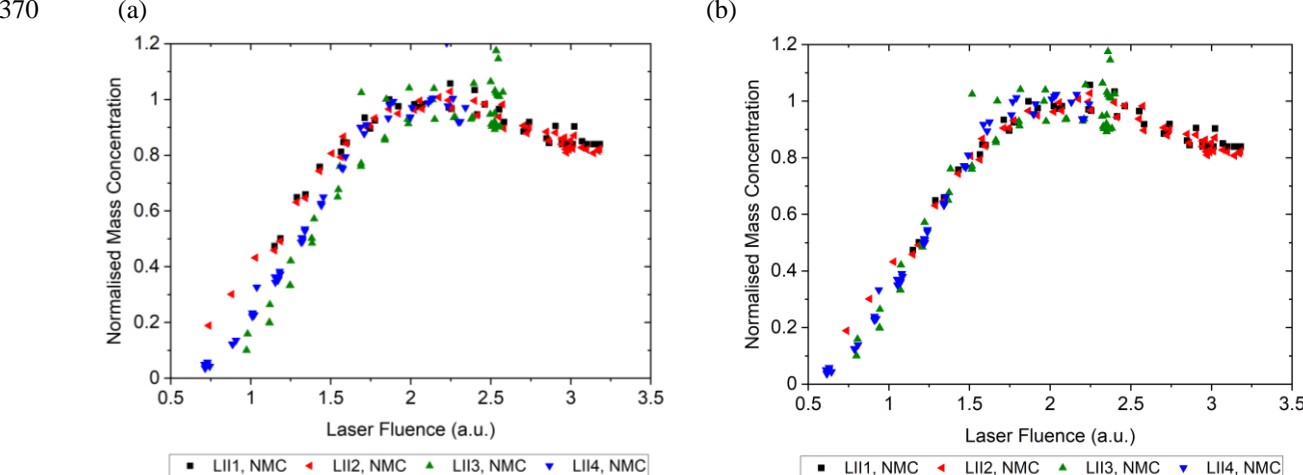

**Fig.6.** Normalised mass concentration (NMC) mean across the different laser fluence measured from the kerosene fueled Rig A by multiple
LII instruments (LIIs 1 to 4): (a) original output from the multiple LII instrument where laser fluence were nominal values, and (b) the laser
fluence axis of instruments LII 3 and LII 4 was shifted to fit the LII 1.


### 3.2.4 Laser fluence – Q-switch delay mapping

During the laser fluence sweep tests, the laser fluence levels were adjusted by changing the Q-switch delay settings. The
correlations in between the Q-switch delay settings and the nominal laser fluence levels from the multiple LII instruments are
different due to modest variations in the characteristics of each laser used in the LII 300 instruments. The correlation maps
obtained during the laser fluence sweep tests at the HPO condition of Rig A are shown in Fig. 7. Each point indicates an
average fluence over a period of 15 seconds duration. Raw data is shown here with no normalisation or adjustments included.
A fifth order polynomial function was fit to the data from each instrument and shown as a solid line in Fig. 7. The fit was
essentially linear except at the lowest Q-switch delay settings, where there was evidence of saturation observed for the four
different LII 300 instruments. LII 1 was also used for measurements from Rigs C to F. For the test on Rig B (IP rig) the LII 1
was not available, and LII 2 was used instead. Note that LII 1 and LII 2 have a very similar relationship between laser fluence
and normalised mass concentration as shown in Fig. 6.





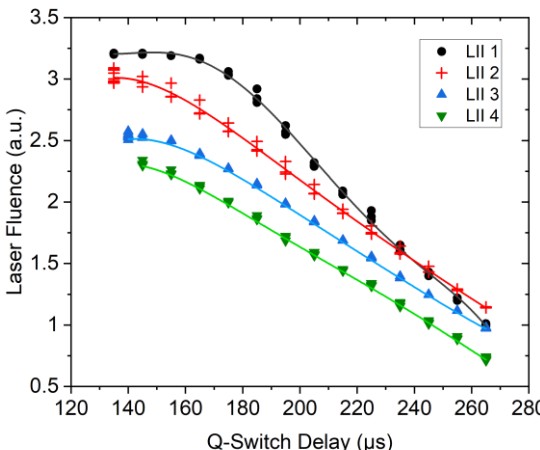

**Fig. 7.** Correlation between the Q-switch delay settings and the laser fluence levels (nominal) from the LII 300 instruments (LIIs 1 - 4).

### 3.2.5 Particle size distribution


Figure 8(a) shows the particle size distributions in electrical mobility diameter ($d_m$) measured using the SMPS in this study from multiple sources as a reference. For spherical particles, $d_m$ is equivalent to the volume equivalent diameter $d_{ve}$; whereas for aggregate particles, $d_m$ is larger than $d_{ve}$ (Decarlo et al., 2004). In Figure 8(b, c), example images of the typical soot morphology collected in a previous study (Saffaripour et al., 2017) from a turboshaft engine exhaust are also shown for

reference. The TEM images from two conditions are shown here: condition E1-1, speed 13,000 rpm, load 70 shaft horse power (shp), and the estimated global equivalence ratio $\varphi$ of 0.25; and condition E1-2, speed 21,000 rpm, load 630 shp, and $\varphi$ of 0.18. The image shows the primary particles and the fractal shaped soot aggregate analysed via transmission electron microscope (TEM). In general, the primary particle diameters were small (17.2 nm (condition E1-1) and 20.8 nm (condition E1-2)), as were the aggregates, and they were nonvolatile, with no evidence of significant semi-volatile coatings (clear

boundaries) (Saffaripour et al., 2017). The aggregate sizes that were determined by projected equivalent-area diameter from TEM images were 33.5 nm (condition E1-1) and 45.7 nm (condition E1-2) (Saffaripour et al., 2017). The geometric mean diameter determined from the particle size distributions from Rig A is in close agreement with the prior results of aggregate diameter from TEM image analysis by Saffaripour et al., 2017.



(a)

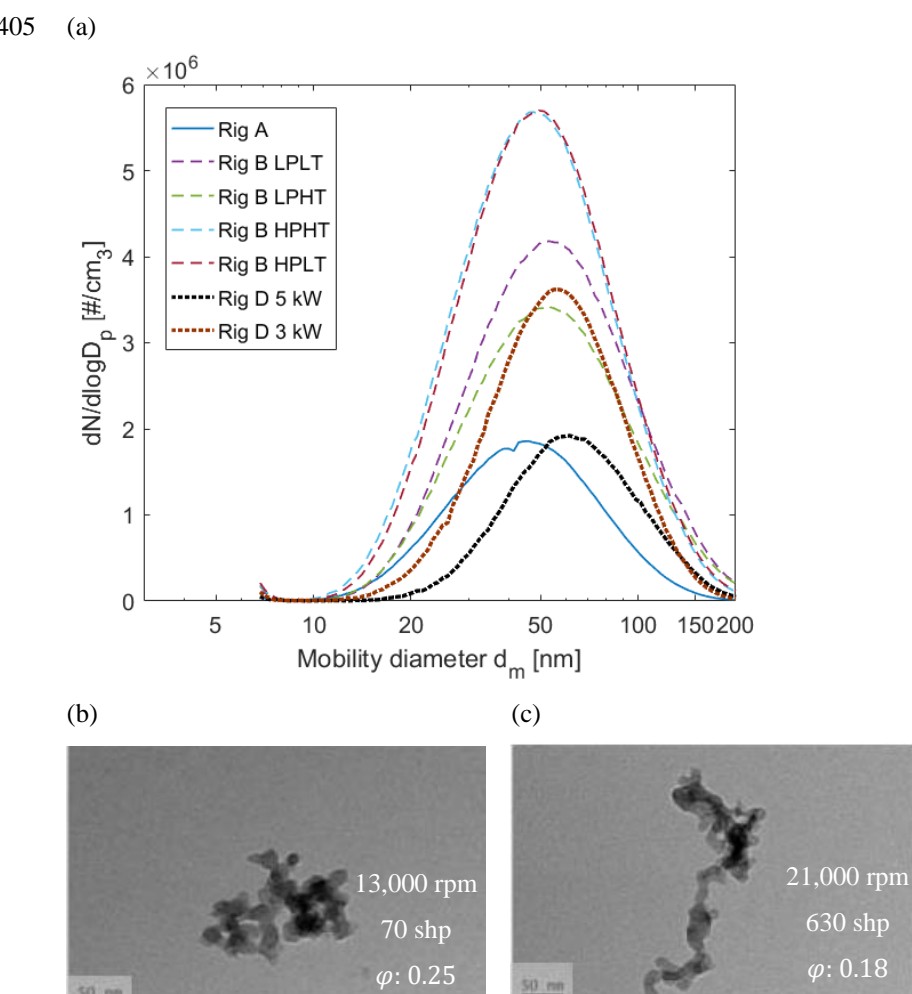

(b)                                    (c)

**Fig. 8.** (a) Typical particle size distributions measured using the SMPS for the multiple sources and operating conditions: Rig A (kerosene fuelled, HPO condition), Rig B (kerosene fuelled, four operating conditions) and Rig D (diesel fuelled, two load conditions); (b, c) typical TEM images of soot produced by a turboshaft engine (speed: 13,000 (b) and 21,000 (c) rpm, shaft horse power: 70 (b) and 630 (c) shp, and the estimated global equivalence ratio of 0.25 (b) and 0.18 (c), respectively) at a magnification of 45,000 times (Saffaripour et al., 2017).

**3.3 LII 300 fluence sweeps from multiple kerosene fueled rigs – Rigs B and C**

Figures 9 and 10 compare the mass concentration profiles as a function of laser fluence levels from the different test rigs, Rig B – IP rig and Rig C – APU, respectively, both operating with kerosene fuel. The laser fluence sweep tests for the four operation conditions of Rig B (Fig.9) resulted in similar trends with fluence for the NMC. In general, the mass concentration results were less dependent on the laser fluence levels (plateau) over the fluence range of 1.8 to 2.5 (nominal values), where peak mass



concentration was observed ~ 2.1 for the two low-pressure cases and ~ 2.2 for the two high-pressure cases. For the NMC at the low-pressure (LP) condition, both the low-temperature (LT) and high-temperature (HT) cases were coincident, however, the two high-pressure (HP) cases were shifted to higher fluence, with the high-pressure low-temperature (HPLT) results having the greatest shift. Sublimation was observed above a fluence of 2.5 (nominal values) for all four test conditions, with the HPLT condition requiring slightly more fluence than the other conditions. The TOA measurements for the four cases indicate similar

organic carbon content, with the OC/TC ratio of 0.43 (LPLT), 0.41 (LPHT), 0.43 (HPHT) and 0.47 (HPLT). The relative high OC/TC ratio suggests that additional laser energy was used to evaporate volatile organic material coating the nvPM. The higher OC/TC for the HPLT condition compared to the rest of the conditions is consistent with requiring additional fluence across the entire range, from low fluence to sublimation.

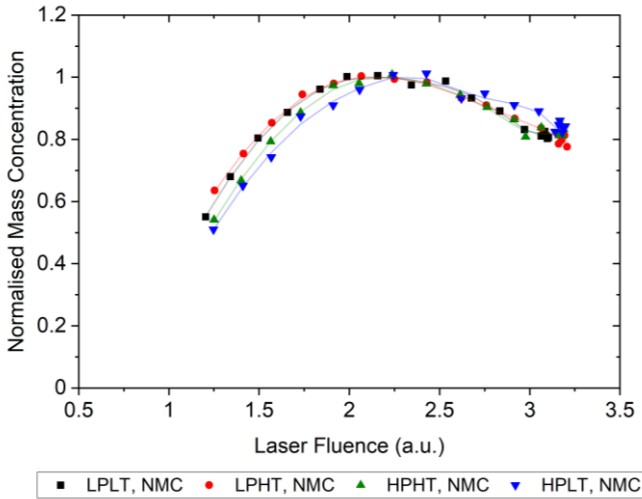

**Fig. 9.** Laser fluence sweep performance using Rig B as the nvPM source, with four operating conditions. NMC – normalised mass concentration. Superimposed are the best-fit curves from the Loess method.





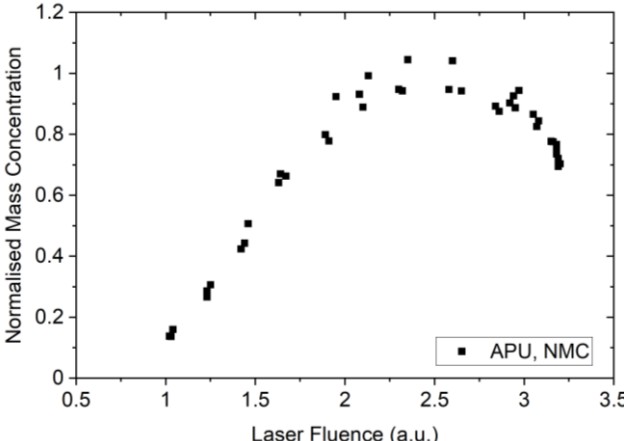

**Fig. 10.** Laser fluence sweep results using sources from Rig C. NMC – normalised mass concentration.


The laser fluence sweep measurements using the nvPM source from Rig C are shown in Fig. 10. Peak mass concentration was obtained at ~ 2.25 laser fluence (nominal value) of Rig C, similar to that from Rig A (~ 2.25) and Rig B (2.2 – 2.5), operating with the same fuel (kerosene). Mass concentration measurements were less dependent on the laser fluences in the range of 2 to 3 (a.u.) for the Rig C.

The similar performance of the LII 300 measurements from multiple rigs demonstrated the robustness of this technique and revealed that an optimised laser fluence can be valid for real-time measurements from a variety of sources. On the other hand, differences shown in the laser fluence shifts required to align the fluence sweeps for different operating conditions, even from the same sources (such as shown in Figs 4-5 from Rig A), suggest that possible differences in the composition (such as volatile organic coatings), morphological characteristics (internal structure of the primary particles or aggregation), or optical

absorption properties of the particles at different operating conditions may be important. This observation suggests that care must be taken in selecting the optimum laser fluence when using a single or different source (such as a laboratory flame) for calibrating LII 300 instruments, such that it is valid (in the plateau regime) for both the calibration source and the intended application.

**3.4 LII 300 fluence dependence –different sources and fuels**

The laser fluence dependence was further investigated for Rigs D – F, which were reciprocating engine sources with intermittent combustion and burned diesel or gasoline fuels, unlike Rigs A-C which were steady-state combustion sources with kerosene fuel. The results are shown in Fig. 11(a) and demonstrate that, similar to the results for Rigs A-C, there is a wide range of laser fluence levels (1.7 - 2.7) where the measurements of mass concentrations were insensitive to the laser fluence



levels (plateau regime). This allows a moderate laser fluence to be applied in practical applications to avoid the mass loss due

to sublimation and at the same time reduce the reliance on a critical laser fluence value, as the same result is obtained across

this range of laser fluences.  Mass concentrations obtained from Rigs D – F were normalised by the maximum of each rig

individually and plotted together with those from Rig A and Rig C in Fig. 11(b), with all the data acquired using the same

instrument (LII 1). As with Rigs A-C, it should be noted that at low fluence levels (<1.7 nominal values for Rigs D-F), the

reported mass concentrations were lower than those measured when the laser fluence level is in the plateau regime.

(a)                                                                                                    (b)

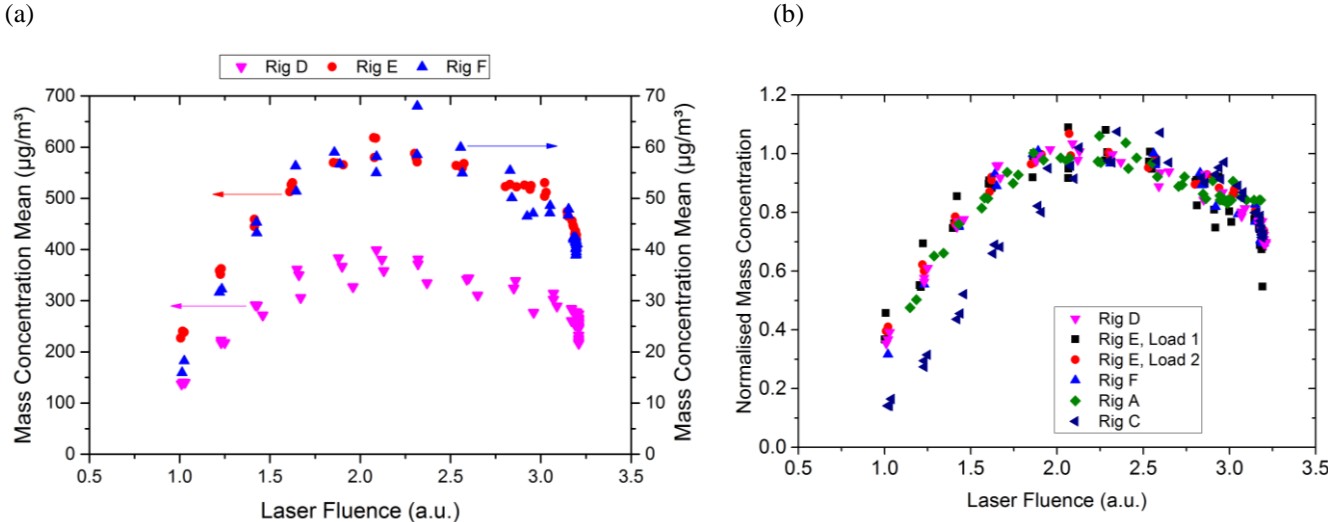

**Fig. 11.** (a) Laser fluence dependence of mass concentration measurements from LII 300 using multiple nvPM sources from diesel (Rigs D
and E Load 2) and gasoline (Rig F) fuelled engine exhausts at typical operation conditions. (Arrows point at the corresponding Y-axis.) (b)
Normalised mass concentration vs laser fluences from Rigs A and C to F. Measurements were carried out by the same instrument LII 1 for
the kerosene cases (Rigs A, C) as for the diesel and gasoline cases (Rigs D to F). Rig E, Load 1: speed 1200 rpm, load 165 Nm and Load 2:
speed 2200 rpm, load 300 Nm.

In general, similar behaviour is observed in the normalised data from all the rigs and fuels in the plateau ranges. Fig. 11(b)

illustrates that the optimum fluence from Rig A overlaps well with that from other rigs, except for Rig C. The results from Rig

C require a shift on the fluence axis (of around 0.3, nominal value) in order to align with those from Rig A (HPO condition).

This shift was due to the combustion formed soot particles emitted from Rig C which had different physical and chemical

characteristics compared to the soot formed from Rig A. Further analysis would require much greater understanding of the

soot morphology, structure, and composition characteristics from the various operation conditions, which is beyond the scope

of this paper.

The laser fluence sweeps showed similar results in terms of the achieving an optimum fluence range with a plateau regime

for the NMC measurements across multiple rigs and fuels, which may suggest the option of utilising cost-effective rigs as

nvPM sources for calibration prior to measurements on aircraft gas turbine engines. The data suggests a near-universality of





the fluence sweeps, with the need to shift some by a fixed amount of fluence to compensate for differences in the particle properties, but all with the same shape and exhibiting a plateau regime over which the response is uniform for a range of laser fluence values. Utilising an optimum fluence level of 2.2 (nominal), which corresponds to the peak of the best-fit from Rigs D to F, will fall within the 2% error band which defines the plateau regime for the HPO condition of the Rig A. It should be noted that the shift in the fluence levels measured on Rig A (shown in Fig. 4) required to align the fluence sweeps at the idle

and HPO conditions (shown in Fig. 5(b)) would lead to a 4% bias error for the idle condition of the Rig A if the fluence is set at 2.2. At engine idle or at conditions where less mature soot (Migliorini et al. 2011) or volatile coatings are encountered, the laser fluence range for the plateau regime (with a 2% error band) will need to be shifted to a higher fluence level, covering the range from 2.4 to 3.2 (nominal values). The optimum fluence to cover the full range of operating conditions for Rig A is the region where these two plateau regime ranges overlap in the range of 2.4 to 2.5 (nominal values).

**3.5 Comparison of mass concentration results from different diagnostic techniques**

The nvPM mass concentration was simultaneously measured using LII 300, PAX, and MSS for a range of conditions for Rigs A-F. Here the real-time nvPM mass concentrations measurements from Rig A are discussed (Fig. 12) as an example for interpreting the results from the laser-induced incandescence and photoacoustic real-time diagnostic techniques.

(a)                                                          (b)

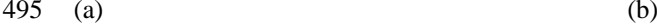

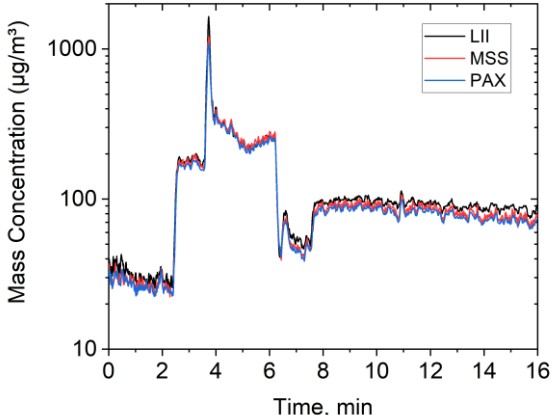
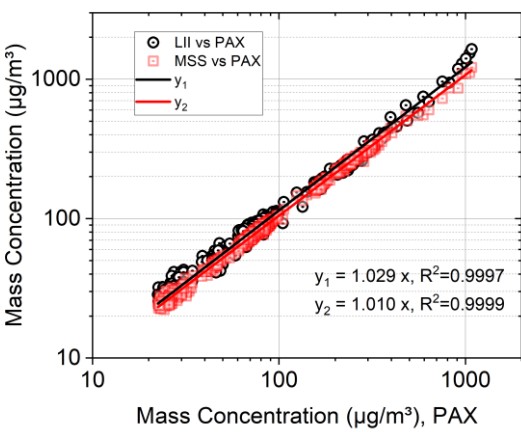

**Fig. 12.** (a) Mass concentration profiles for LII 300, PAX, and MSS and, (b) correlations between LII 300 and MSS nvPM mass concentrations to those from PAX (Rig A).

Figure 12 shows that agreement high degree of correlation was found between the LII 300, PAX, and MSS for nvPM mass concentration measurements. From the linear curve fits over nearly two orders of magnitude ($y_1$ – LII vs PAX and $y_2$ – MSS vs PAX, with an intercept value 0), the mass concentrations from the LII 300 and PAX are within 2.9% of each other, and the





mass concentrations from the LII 300 and MSS are within 1.9% of each other, all much less than the uncertainties associated with the instruments.

The LII 300 measurements for nvPM mass concentrations were further compared with the EC results obtained from TOA. The results from the multiple rigs are shown in Fig. 13 normalised to the EC results. The measurements from the PAX and MSS instruments are also included for comparison. The nvPM mass concentrations measured by the real-time instruments, i.e., LII 300, PAX, and MSS, were averaged for the same duration as the filter collection by the corresponding TOA measurements. The error bars of real-time instruments were computed by calculating the relative standard deviation, RSD

(standard deviation of the sampled mass concentration data normalised to its average value), indicating the variability in the nvPM mass measurement. The LII 300, PAX, and MSS results generally are not significantly different than those for EC from TOA, with error bars overlapping within the uncertainty (16.7%) of the TOA EC determination. The real-time instruments' results were higher (1~20%) than the EC results from TOA on nvPM emissions from most rigs, except the PAX result from Rig D - 3 kW. The PAX result from Rig D – 3 kW was 4% lower than that for the EC result, although was still within the

uncertainty of the TOA measurement. Unlike other cases investigated where the nvPM mass concentrations were observed to be in the range of 100 – 800 µg/m³, the nvPM mass emissions of Rig F were lower, ~ 40 µg/m³. The TOA's uncertainty from the NIOSH method is greatest at lowest mass concentrations (NIOSH 2003), and is reduced at higher mass concentrations. In comparison of the real-time instruments' performance, for the cases investigated, PAX results were on average 6% lower than that for the LII 300 from the kerosene fuelled rigs, 10% lower from the diesel fuelled rigs, and 6% lower PAX results compared

with the LII 300 from the gasoline fuelled rig. MSS results were on average 7% higher than that for the PAX from both kerosene fuelled and diesel fuelled rigs. This agrees in general with the finding of previous studies (Smallwood et al. 2010; Durdina et al., 2016; Lobo et al., 2020; Corbin et al., 2020), and the discrepancy between the various instruments from source to source and amongst the different fuels is likely caused by difference in the properties of the particles (morphology, structure, and optical absorption), the different content of non-refractory components of the particles (i.e. quantities of bound H and O,

volatiles, nitrates, ash, sulphates, etc.) (Smallwood et al., 2010), and uncertainties associated with varying relative humidity content in the heated sampling cell (for photoacoustic instruments) (Arnott, 2003), as well as the choice of the split point in determining EC from TOA (Baumgardner et al., 2012). In terms of identifying a substitute for the aircraft gas turbine helicopter engine (Rig A) for calibrating the LII 300, Rig C (APU) appears to be the closest in terms of LII 300 response, with 1% higher than EC from TOA on the same source, well within the uncertainty of the methods. In terms of the variability in nvPM mass

concentrations results, the LII 300 exhibited a lower variation of 5% on average, among all the rigs and operating conditions investigated, compared to that for the other instruments, which was 6% for PAX, 7% for MSS, and 7% for EC from TOA. A higher variability was shown at cases of low nvPM mass emissions (such as from Rig F or Rig D at 3kW, refer to nvPM mass concentrations' ranges detail in Fig. 13 caption) for both PAX and MSS than the variability exhibited at cases of high nvPM mass emissions (such as in Rigs B and E). This trend of a low variability in nvPM mass at high concentrations is consistent

with results from emissions measurements of miniCAST soot generator (Lobo et al., 2020) and other aircraft engines (Lobo et al., 2015b, 2016).

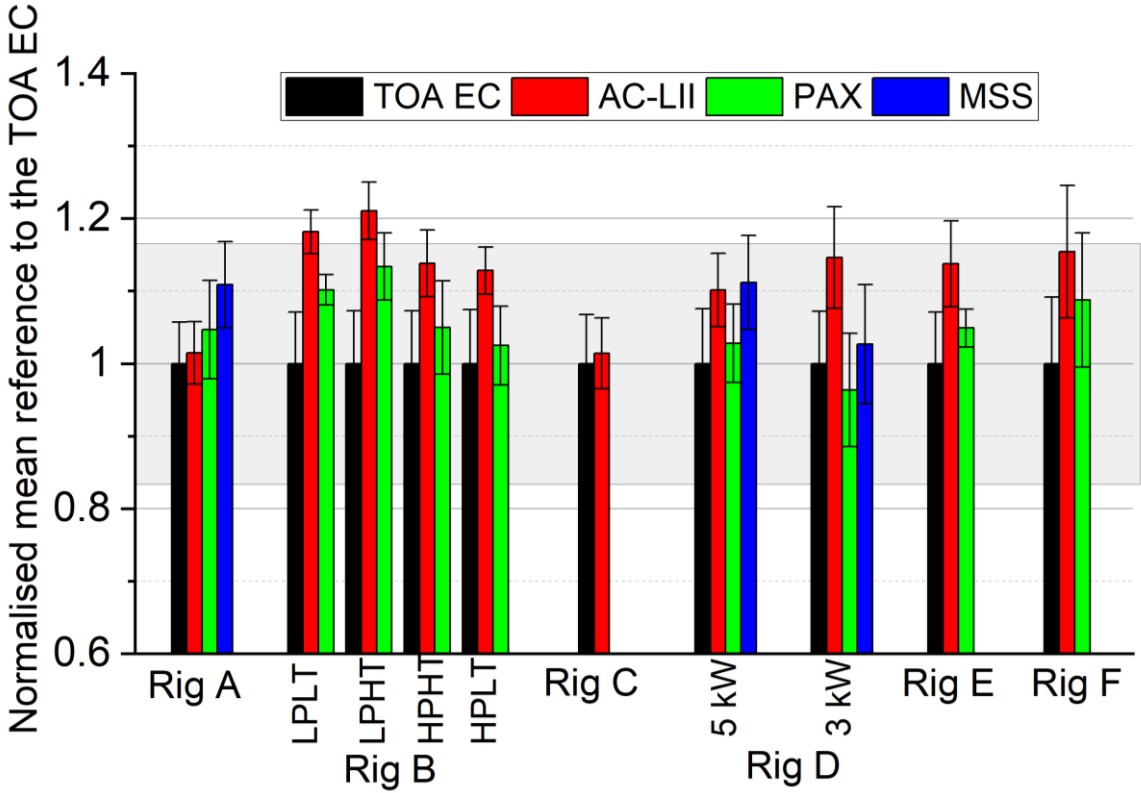

**Fig. 13**. Comparison of the mass concentration results from the multiple rigs and diagnostic techniques. The shaded area marks the 16.7% uncertainty of the TOA from the NIOSH method (NIOSH 2003). The nvPM mass concentrations are in the range of 90 – 470 µg/m³ from Rig A, 500 – 600 µg/m³ from Rig B, 670 - 780 µg/m³ from Rig C, 300 – 550 µg/m³ from Rig D 5 kW, 200 – 270 µg/m³ from Rig D 3 kW, about 590 µg/m³ from Rig E, and about 40 µg/m³ from Rig F.

**Summary**

New standards and recommended practices adopted by ICAO require the nvPM mass and number emissions from aircraft engines to be quantified during emissions certification tests. The LII 300, the only commercial LII instrument utilising the AC-LII method, has been used to measure nvPM mass emissions from aircraft engines. Previous studies reported the sensitivity of the LII technique to the type of black carbon sources. In this study the response of the LII 300 instrument to different nvPM from a range of different sources and fuels was investigated to understand the relationship between the laser fluence values and the resulting nvPM mass concentrations, and to evaluate the suitability of different sources/fuels to be used as an nvPM calibration source for the LII 300. For all the tests using multiple rigs as sources of nvPM emissions, LII 300 measurements demonstrated a plateau regime with a range of laser fluence values where the resulting nvPM mass concentration measurements were insensitive to the laser fluence levels applied. The shape of the fluence sweep curves was nearly universal for all sources, operating conditions, and fuels investigated. Optimising the laser fluence for the plateau regime over the range of source





operating conditions was shown to reduce potential uncertainties for the LII 300 associated with the corresponding range of nvPM properties.

555       Evidence demonstrated that the LII 300, PAX, and MSS had similar response and performance in the real-time measurements of nvPM emissions from multiple rigs studied. Compared to other diagnostic instruments, the real-time measurement output of LII 300 exhibited no significant differences and high correlation (>97%) with the photoacoustic instruments. To assess suitability of replacing an aircraft gas turbine engine as a calibration source, further work is required to establish the repeatability and reproducibility of particles sources, as well as investigating additional laboratory sources

including the miniCAST, MISG (mini-inverted soot generator), and nebulized carbon black particles. In addition, future work should include investigating the morphology characteristics, composition, and optical absorption of the various particulate matter sources from multiple operating conditions, to further understand the relationship between soot particle characteristics and the response of real-time instruments used for the measuring nvPM mass concentration.

**Data availability**

The datasets analysed during the current study are available from the corresponding author on request.

**Author contributions**

G.J.S., M.P.J., P.L., M.C.P., and A.S. conceived and planned the study, and performed the experiments. R.Y. processed the

experimental data and performed the analysis. R.Y. drafted the manuscript with input from P.L. and G.J.S., and designed the figures. P.L. and G.J.S. assisted in interpreting the results. D.B. contributed to the discussion on the time-weighted normalisation method. All authors discussed the results and contributed to the final manuscript.

**Competing interests**


The authors declare that they have no conflict of interest.

**Acknowledgements**

The authors would like to acknowledge the funding from Transport Canada for this project, and support from Rolls-Royce. The authors also thank Dan Clavel and Brett Smith who assisted with the data collection. Ruoyang Yuan would like to

acknowledge the funding from the EPSRC and the David Clarke Fellowship (EP/S017259/1) to support her work.



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
