# Peer review of "Measurement of Black Carbon Emissions from Multiple Engine and Source Types using Laser-Induced Incandescence: Sensitivity to Laser Fluence"

_Atmospheric Measurement Techniques, 2021_

## Author Comment (AC4)

[Figure]

**Fig. 11.** (a) Laser fluence dependence of mass concentration measurements from LII 300 using multiple nvPM sources from diesel (Rigs D and E Load 2) and gasoline (Rig F) fuelled engine exhausts at typical operation conditions. Superimposed are the best-fit curves from the Loess method. (Arrows point at the corresponding Y-axis.) (b) Normalised mass concentration vs laser fluences from Rigs A and C to F. Measurements were carried out by the same instrument LII 1 for the kerosene cases (Rigs A, C) as for the diesel and gasoline cases (Rigs D to F). Rig E, Load 1: speed 1200 rpm, load 165 Nm and Load 2: speed 2200 rpm, load 300 Nm. (c) Combined data from Rig A (idle) and Rig B (at four conditions shown in Fig. 9b) to (b), the laser fluence axis of Rig C and Rig E, Load 1 was left-shifted by 0.27 and - 0.05 mJ/mm², respectively.

In general, similar behaviour is observed in the normalised data from all the rigs and fuels in the plateau ranges. Fig. 11(b) illustrates that the optimum fluence from Rig A overlaps well with that from other rigs, except for Rig C. The results from Rig C require a shift on the fluence axis (of around 0.3 mJ/m²) in order to align with those from Rig A (HPO condition), as shown in Fig. 9c where the fluence data from six rigs and a total of 11 conditions converge after the fluence shifts were applied. This shift was due to the combustion formed soot particles emitted from Rig C which had different physical and chemical characteristics compared to the soot formed from Rig A. Further analysis would require much greater understanding of the soot morphology, structure, and composition characteristics from the various operation conditions, which is beyond the scope of this paper.

---

## Author Response (AR1)

We thank the reviewers for the review of the manuscript and supportive comments. The response to each reviewer comment is provided below.

Colour code: Black – reviewer comments; blue – author response; red – corresponding line number of the changes.

RC1:

**General comments:**

1. The LII community also uses the term nrBC (non-refractory black carbon), for example in atmospheric studies using the SP2 instrument. To allow for a better understanding between different communities, it would be valuable to discuss the terminology relations (nvPM vs. nrBC, for example) in addition to "soot" and "black carbon" already in the introduction. It could be also valuable to mention/discuss the SP2 approach, expected similarities, and differences in mass measurements, for example, etc.

   This paper we focused on the particular application for aircraft engine. The authors feel it may introduce confusion if using other terms (such as refractory Black Carbon (rBC) or single-particle soot photometer (SP2) instrument used in atmospheric studies).

2. Maybe I missed it, but how is the fluence measured? The graphs are in a.u., and the authors talk about nominal fluence. I guess this is the fluence measured by some photodiode in the instrument itself. Is that correct? Is there any information provided on the calibration (linearity and slope) of these fluence values?

   The fluence was measured from an energy meter in the LII instrument and laser beam cross-section area in the probe volume, obtained from a beam profile camera. The information is added in the revised manuscript, in L272-274 and L364-367. The arbitrary unit of the laser fluence was changed to the measured value in mJ/mm$^2$ in the text and figures (Fig. 2c, Figs. 4-7, and Figs. 9-11).

   Figure 7 shows that the relationship between fluence and q-switch delay is consistent from instrument to instrument and lends confidence to the determination of laser fluence in this work.

3. I think a summary table with the optimal fluence ranges for the different sources would help in the future to provide a quick view of best operational conditions.

   Optimal fluence ranges were clarified in the text for the different sources. Although with laser fluences shifting, these optimal range can collapse, the actual values (fluence shifting or the original optimal fluence range) may differ from

source to source and be affected by the soot particle properties. Therefore, a summary table would not add additional value to the manuscript.

**Specific comments**

1. Lines 23-25: I find the following sentence a bit confusing, if an optimized and therefore constant (?) fluence is used, why would different fluence levels be used? Maybe the authors mean that in a range of fluence around the optimized vale, the mass concentration is unchanged? "It was found that an optimised laser fluence can be valid for real-time measurements from a variety of sources, where the mass concentration was independent of laser fluence levels covering the typical operating ranges for the various sources."

   Sorry for the confusion. A constant (optimal) fluence is used in the real-time engine test.

   To obtain this optimal fluence value, a series laser fluence sweep tests need to be performed covering the range of operation conditions of the source, such as demonstrated in Fig. 4 that the fluence sweep was performed at the idle and the high-power output conditions. The optimum value is determined from the optimum ranges from the sweep tests, as described in L308-312.

2. Lines 190 – 194: Filter-based measurements can also be negatively affected by humidity.

   While it is true that filter-based measurements can be affected by humidity, the exhaust source for the measurements was heavily diluted prior to sampling (see figure 1). We did not observe any negative effects of humidity for the filter samples.

3. Line 198: I guess the PAX operates at 870 nm, but what about the MSS? Also 870 nm? Please clarify.

   The MSS operates at a wavelength of 808 nm. The manuscript has been updated with this information (P7, L203).

4. Line 199: Even with RI constant, the AAE might deviate from 1 somewhat, so the equivalence might not be perfect.

   Since the measurements reported in this manuscript are for non-volatile particulate matter, the assumption of an AEE = 1 is reasonable.

5. Line 200: What source was it? Also, how was the PAX calibrated?

   The calibration source for the MSS was a CAST. The PAX was calibrated separately with ammonium sulphate and Aquadag solutions. The information is added to the revised manuscript in L203 - 205.

6. Lines 258-260: Consider rewording the sentence to something like "A time-weighted normalisation (TN) method was used to account for scatter caused by any modest variations in the concentration of the source emissions". Also, why could one use the SMPS concentrations or the photoacoustic signals to account for source concentration fluctuations?

The sentence has been reworded as suggested. One could use data from another measurement such as SMPS, PAX to account the source concentration fluctuations. In Fig. 12a, it shows the good response and agreement of PAX, MSS and LII to the fluctuation in the real-time measurement. But for the current study, the focus is on fluence characteristics from the LII instruments, especially in the fluence sweep characteristics, we weren't look at other instruments behaviours to account for the fluctuations.

RC2:

**Points should be improved:**

1. The MSS is known to be influenced from ambient effects, like changes in humidity. I suspect a similar behavior of the LII, and I am convinced the manuscript would benefit from an additional discussion of ambient effects on LII measurements. E.g., what were ambient conditions during the measurements?

In many investigations, we have never observed effects of humidity on the mass concentration reported by the LII 300 instrument. The instrument does measure the pressure and temperature in the sample cell and corrects the mass concentration to that at STP conditions of 0 °C and 1 atm. In other words, the ambient conditions do not have an impact on the measurement.

LII is not an absorption-based instrument, and at the temperatures the particles are heated to (~4000 K) the ambient conditions do not have an effect on the measurement of mass concentration. MSS is an absorption-based instrument, which only perturbs the temperature slightly from ambient conditions, and may be more susceptible to temperature and humidity.

2. The authors performed the measurements on the different rigs over different periods of time: Other than in Fig. 3, the reader gets very little information how the authors made sure to assume "stable combustion conditions" for all rigs. Was there any CO2 measurement attached to the rig? Is there any EGT measurement available, which could be used as a potential tracer for combustion stability? As well, little is known about the warm-up sequence of the engines, neither do the author describe if any exhaust gas treatment (especially for rigs E & F) was present.

In terms of the warm-up and stabilising, the engines/rigs were running at the set point for a short period while the real-time data of temperature and other operating conditions were monitored. Once these operational parameters were determined to be stable, the data collection for that particular set point was initiated.  In terms of the EGT, a thermocouple was fitted on the exhaust of each engine/rig. The temperature along with other data were monitored and available as a tracer for combustion stability.  There were no exhaust after-treatment on any of the engines/rigs.

3. What is the essence of the project? What are, after all the measurements performed by the authors, the recommendations? Can a low-cost engine like in rig F be used as a calibration device for an LII, if aircraft emission measurements following ICAO Annex 16 Vol. 2 be performed operationally? If yes, how large are the remaining uncertainties in terms ov nvPM mass, and how does this uncertainty compare to e.g. MSS measurements?

In section 3.5, P24, L546, we discussed that 'In terms of identifying a substitute for the aircraft gas turbine helicopter engine (Rig A) for calibrating the LII 300, Rig C (APU) appears to be the closest in terms of LII 300 response, with 1% higher than EC from TOA on the same source, well within the uncertainty of the methods'. The uncertainties were shown in Fig. 13 for each rig and addressed in P25, the second paragraph.

From the rigs investigated in this study, the APU seems to be the closet low-cost alternative engine calibration source for an LII. The next candidate would be the diesel generator but operating at a high power output (i.e., in the cases studied, Rig D, at 5 kW). The mass concentration results are within 1% between LII and MSS, and are about 10% higher than the TOA EC results. The error bar of each instrument (LII, MSS, PAX) overlaps within the uncertainty (16.7%) of the TOA EC determination. However, to assess suitability of replacing an aircraft gas turbine engine as a calibration source, 'further work is required to establish the repeatability and reproducibility of particles sources, as well as investigating additional laboratory sources including the miniCAST, MISG (mini-inverted soot generator), and nebulized carbon black particles', addressed in the summary section.

**Specific Comments**

1. L165: Add information what total volume was sampled onto the quarz filter

Information on the total volume sampled on to the quartz filter, along with other measurement details, has now been included in the manuscript (L165-169).

2. Methods: How did you define "stabilized conditions" for representative measurements? How long did you wait after any load change on the engines? Were the engines warmed up? Was any of the reciprocal engines fitted with any exhaust treatment mechanism?

In terms of the warm-up and stabilising, the engines/rigs were running at the set point for a short period while the real-time data of temperature and other operating conditions were monitored. Once these operational parameters were determined to be stable, the data collection for that particular set point was initiated.  In terms of the EGT, a thermocouple was fitted on the exhaust of each engine/rig. The temperature along with other data were monitored and available as a tracer for combustion stability.  There were no exhaust after-treatment on any of the engines/rigs.

3. Fig. 2/4 & others: It is only explained in line 358/359 why you are using arbitrary units instead of mJ/mm^2. I suggest adding this information earlier in your manuscript.

The arbitrary units were replaced with mJ/mm^2 in the graphs and texts in the revised manuscript. We added a note on how the fluence was determined in the text, in P11, L272-274, and P15, L364-367.

2. Fig. 10/11: I recommend adding a Loess curve as in Fig. 9 for consistency
   The Loess curve are added in Fig. 10 (P20) and Fig. 11 (P21) as suggested.

3. Fig. 13: I am missing an explanation why the range for Rig E & F can't be more specifically indicated ("uncertainty" or "variability"?)
The range for Rig E and F has now been added in the caption of Fig. 13 (P25, L560).

**Technical Corrections**

1. Added 'The' in L43.
2. L49: Removed CAEP, SARPs abbreviation as suggested.
3. L80: Moved all cited papers at the end of the sentence as recommended.
4. L123: Added the two wavelengths information in the sentence.
5. L141: Added comma after "or EC".
6. L151: Revised the sentence adding 'therefore'.
7. Changed relative to relatively in L434.
8. Rewrote the sentence in L455-L459.
9. This sentence has been reworded (now in L533).

Additional change made:

Added Figure 11c (P21) and the text (L491), showing the fluence data from six rigs and a total of 11 conditions after shifting the fluence axis of Rig C and E, load 1.

List of all relevant changes made in the manuscript:

1. Information on how the laser fluence value was obtained are added in the revised manuscript, in L272-274 and L364-367. The arbitrary unit of the laser fluence is changed to the measured value in mJ/mm$^2$ in the text (such as in L304, 306 P13), and figures (Fig. 2c (P11), Figs. 4-7 (P14-17), and Figs. 9-11 (P19 -21)).

2. The MSS operating wavelength is added to the manuscript (P7, L203).

3. The calibration sources for the MSS and PAX are added to the revised manuscript in L203 - 205.

4. The total volume sampled on to the quartz filter has now been included in the manuscript (L165-169).

5. The Loess curve are added in Fig. 10 (P20) and Fig. 11 (P21) as suggested for consistency.

6. The range for Rig E and F has now been added in the caption of Fig. 13 (P25, L560).

7. All the technical corrections commented by RC2 are revised accordingly. They are:
   1) Added 'The' in L43.
   2) L49: Removed CAEP, SARPs abbreviation as suggested.
   3) L80: Moved all cited papers at the end of the sentence as recommended.
   4) L123: Added the two wavelengths information in the sentence.
   5) L141: Added comma after "or EC".
   6) L151: Revised the sentence adding 'therefore'.
   7) Changed relative to relatively in L434.
   8) Rewrote the sentence in L455-L459.
   9) This sentence has been reworded (now in L533).

8. Added additional sub-figure, Figure 11c (P21), and the text (L491), showing the fluence data from six rigs and a total of 11 conditions after shifting the fluence axis of Rig C and E, load 1.

---

## Referee Report (RR1)

Review of the Manuscript amt-2021-209 titled "*Measurement of Black Carbon Emissions from Multiple Engine and Source Types using Laser-Induced Incandescence: Sensitivity to Laser Fluence*"

The paper investigates the response of the LII 300 instrument to different non-volatile particulate matter (nvPM) emitted by different sources and fuels. In particular, the effect of laser fluence on the nvPM mass concentration was investigated in order to identify a possible nvPM source to be used as calibration standard for the LII 300.
The study was conducted rigorously and the results constitute a valuable contribution in the development of calibration protocols for nvPM mass measurements with the LII 300.

As a general comment, the manuscript is well written and organized. The approach of using the fluence curve to identify on optimal fluence for LII measurements of different type of nvPM is a good practice for LII measurements but it is very often not used and discussed. Therefore, the present paper represents a very valuable study for the identification of a suitable LII calibration source and I recommend the manuscript for publication after few minor revisions.

Point to be improved:

The peak particle temperature presented in Fig. 5 is determined from the two-color pyrometry method and therefore assumptions on the optical properties of the particles have been made.
The fact that the temperature reached form the particles produced at the idle condition is lower to the one obtained by those emitted at the HPO conditions, might be due to the fact that the two type of particles have different optical properties.
The authors pointed out later in the text that differences in particle optical properties might cause a shift in the laser fluence curve, however I believe that a short discussion about the assumption on the optical properties would be valuable in completing the description of figure 5.

Specific comment
L314:for sake of clarity I suggest to add that the effective primary particle diameters are reported in Fig.4

Technical comment
L347: please remove "arbitrary"

---

## Author Response (AR2)

Dear Associate Editor and the editorial support team,

Thank you very much for your review and great support!

We'd like to thank Reviewers for the review of the manuscript and supportive comments. The response to each reviewer comment is provided below.

Colour code: Black – reviewer comments; blue – author response; red – corresponding line number of the changes. Changes to the manuscript are in highlight.

**Report #1 (Referee #1):**

**Comments:**

1. If point 2 of my specific comments (2. Methods: How did you define "stabilized conditions" for representative measurements? How long did you wait after any load change on the engines? Were the engines warmed up? Was any of the reciprocal engines fitted with any exhaust treatment mechanism?) is implemented into the manuscript (definition of stabilized conditions & no exhaust after-treatment), I would be accepting the manuscript without any other changes.

   A paragraph describing the required information is added to the revised manuscript (P8 L238 - 242). The paragraph is copied below:

   > 'To ensure stabilised conditions were reached, the sources (test rigs) were operated at the set point for a short period prior to nvPM data collection. The exhaust temperature, measured with a thermocouple fitted to the exhaust of each rig, along with other operating condition data were monitored and available as an indication of combustion stability. Once these operational parameters were determined to be stable, the data collection for that particular set point was initiated. There were no exhaust aftertreatment devices on any of the rigs in the current work.'

**Report #2 (Referee #4):**

**Comments:**

1. The peak particle temperature presented in Fig. 5 is determined from the two-color pyrometry method and therefore assumptions on the optical properties of the particles have been made.
   The fact that the temperature reached form the particles produced at the idle condition is lower to the one obtained by those emitted at the HPO conditions, might be due to the fact that the two type of particles have different optical properties.

The authors pointed out later in the text that differences in particle optical properties might cause a shift in the laser fluence curve, however I believe that a short discussion about the assumption on the optical properties would be valuable in completing the description of figure 5.

The reviewer is correct. The temperature is determined by two-colour pyrometry, and therefore influenced by the optical properties. The absolute value of the absorption function, E(m), is not important in determining the temperature. However, the relative value of E(m) between the two detection wavelengths is important. The assumption being used in this study is that the value is the same at both wavelengths (Snelling et al., 2005). To compare the effect of E(m) varying with wavelength to constant E(m), Snelling et al. (2004) showed that a varying E(m) would result in a peak temperature 80 K greater than using a constant E(m). Changing the approach used for E(m) will have an effect, and potentially the approach used and absolute values will be different at the idle and high power conditions, and this could account for <100 K of the ~350K differences in the peak temperature observed in Figure 5(a).

In line with reviewer's suggestion, a short discussion on above is added to the revised manuscript (P14, footnote) stating "The temperature is determined by two-colour pyrometry, and therefore influenced by the nvPM optical properties, i.e. the relative value of E(m) between the two detection wavelengths. In this study, it is assumed that the value of the absorption function, E(m), is the same at both wavelengths (Snelling et al., 2005). The different particle properties at HPO and idle may invalidate this assumption, but the potential effect would only account for <100 K (Snelling, 2004) of the observed difference in the peak temperatures."

2.  L314:for sake of clarity I suggest to add that the effective primary particle diameters are reported in Fig.4
    The sentence is revised to add clarity (P13, L318). It now reads as:
    'While not the focus of this study, it is interesting to note the impact of laser fluence and source operating condition on the effective primary particle diameters (ePPD) resulting from the LII 300 measurements as shown in Fig.4 (via the decay rate of the LII signal) (Schulz, 2006)'.

3.  L347: please remove "arbitrary".
    It is removed (P15, L351).

**Additional changes made to the manuscript:**

Throughout the text, the fluence is shown as $mJ/m^2$, but it should be $mJ/mm^2$. This unit error has been corrected (for example, in P13, L308-310).